Effects of combined application of compound fertilizer and biochar on absorption and utilization of phosphorus by wheat

Li Yun 1
Wang Jiatong 2
Lu Jiahui 2
Feng Yanfei 2
Li Jingjing 2
Guo Zhenqing 2
Lin Xiaohu xiaohulin2008@163.com 2
Han Yucui yucuihan84@163.com 2
1 Research Center of Rural Vitalization, Hebei Normal University of Science and Technology , Qinhuangdao , China
2 Hebei key Laboratory of Crop Stress Biology, Hebei Normal University of Science and Technology , Qinhuangdao , China
Pandey Renu
Electronic publication date: 2025 Nov 11
Publication date: 2025
Volume: 13
Electronic Location ID: e20308
Received 2025 May 2; Accepted 2025 Oct 8
Copyright: ©2025 Li et al.
Copyright year: 2025
Copyright holder: Li et al.
License: This is an open access article distributed under the terms of the Creative Commons Attribution License, which permits unrestricted use, distribution, reproduction and adaptation in any medium and for any purpose provided that it is properly attributed. For attribution, the original author(s), title, publication source (PeerJ) and either DOI or URL of the article must be cited.
License URL: https://creativecommons.org/licenses/by/4.0/

Keywords: Wheat, Phosphorus, Soil, Absorption, Utilization

Funding: Hebei Province Agricultural Science and Technology Achievements Transformation Fund 2024V1705570221728 This work was supported by the Hebei Province Agricultural Science and Technology Achievements Transformation Fund project (2024V1705570221728). The funders had no role in study design, data collection and analysis, decision to publish, or preparation of the manuscript.

==============================
Unreasonable fertilization practices are still applied in agricultural production, resulting in reduced crop yields and wasting resources. Therefore, it is essential to optimize fertilization regimes to improve the nutrient absorption capacity of crops. However, the effects of combining nitrogen, phosphorus, and potassium (NPK) compound fertilizer with biochar on the uptake and utilization of phosphorus (P) by wheat and soil fertility remain poorly understood. Thus, we tested three biochar levels (0 quintal (q) ha−1 (B1), 75 q ha−1 (B2), and 150 q ha−1(B3)) and four compound fertilizer levels (0 q ha−1 (F1), 2.25 q ha –1(F2), 4.50 q ha−1 (F3), and 6.75 q ha−1 (F4)), and compared the effects of different experimental combinations across two years. The results showed that combined application could increase the wheat yield, soil P content, accumulated plant P, and P utilization efficiency. Among the different treatments, the wheat yields in both years were highest under B2F3, and the total soil P contents in the booting and flowering stages were also the highest. The soil available P contents in the jointing stage, booting stage, and flowering stage were highest under B3F3. The available P soil contents at maturity in both years were highest under B3F4. In addition, compared with B1F4, the accumulated plant P amounts in the booting and flowering stages were 17.6–77.3% and 17.0–87.3% higher, respectively, under B2F2, B2F3, B3F2, and B3F4 (P < 0.05). The accumulated plant P amounts in the maturity stage during the two years were highest under B2F3 (0.83 q ha−1) and B3F3 (0.42 q ha−1), respectively. Moreover, the partial factor productivity and agronomic use efficiency were significantly higher under B2F2, B2F3, B3F2, and B3F3 than B1F4. In summary, under the experimental conditions in this study, applying compound fertilizer at 4.50 q ha−1 combined with biochar at 75 q ha−1 is recommended as the optimal fertilization scheme.

Introduction

Wheat is the main food crop in northern China and maintaining stable yields is highly important for ensuring food security in China (Ma et al., 2023). Phosphorus (P) is an essential nutrient element for wheat growth and development, and it plays important roles in enhancing the yield and quality of wheat (Yu, Yao & Wang, 2022). However, after its application to soil, P is readily fixed through physical and chemical reactions, resulting in low mobility in the soil, and poor absorption and utilization by plant roots (Liang et al., 2024), thereby decreasing the wheat yield, wasting fertilizer resources, and allowing P to accumulate in the soil, with adverse effects on farmland and water environments (Wang et al., 2023; Zhu et al., 2018). Chinese farmers usually apply excessive amounts of phosphate fertilizer to compensate for the low efficiency utilization of phosphate fertilizer and meet the growing demand for wheat production (Xu et al., 2023). However, the excessive application of phosphate fertilizer tends to increase the costs of agricultural production, as well as exacerbating the waste of resources and environmental risk (Li et al., 2011). Therefore, it is highly important to appropriately adjust fertilization methods to improve the absorption of P and its efficient utilization by wheat plants, and reduce negative environmental impacts (Chang et al., 2024).

Many studies have demonstrated that the appropriate application of nitrogen (N), P, and potassium (K) fertilizers can increase the P absorption amount and rate by wheat plants to promote plant growth and the transport of more P to the grains, thereby improving the fertilizer utilization efficiency and wheat yield (Liu et al., 2024a; Rakshit et al., 2015; Lollato et al., 2019). Biochar has a rich pore structure and large surface area, and thus it is highly effective at adsorption and reducing the loss of nutrients in soil (Khan et al., 2024; Gao et al., 2022). In addition, biochar contains a variety of nutrients that can be utilized to improve the basic fertility of soil and directly supply nutrients required by plants, thereby promoting crop growth (Hossain et al., 2024; Haefele et al., 2011). The P content can be enhanced after applying biochar to soil, as well as reducing the loss of soil P through leaching to promote the absorption of P by crops (Van Zwieten et al., 2010; Feng et al., 2012). For example, Glaser & Lehr (2019) found that applying biochar to acidic and neutral soils could increase the soil available P by 5.1 times and 2.4 times, respectively. Moreover, Gupta et al. (2024) demonstrated that compared with low biochar application rates, high biochar application rates significantly increased the availability of P in both loam and clay loam soils. Previous studies mainly focused on the effects of combining the application of a single fertilizer element with biochar on crops (Li et al., 2020; Parvage et al., 2013). For example, Dong et al. (2021) found that combining nitrogen fertilizer with biochar increased the soil fertility, thereby increasing the wheat yield. Farrell et al. (2014) showed that under the same biochar application level, applying a higher phosphorus fertilizer rate could significantly increase the wheat yield. However, only a few studies have investigated the combined application of compound fertilizer and biochar. In particular, Zhang, Liao & Li (2021) showed that when combined with nitrogen, phosphorus, and potassium (NPK) compound fertilizers, the application of biochar effectively improved the soil environment and enhanced the effectiveness of soil nutrients to promote the growth of lettuce. Peng et al. (2021) found that the combined application of compound fertilizer and biochar (referred to as combined application in the following) could significantly increase the maize yield compared with the addition of compound fertilizer alone.

Therefore, NPK compound fertilizer combined with biochar may promote the absorption and utilization of P in wheat and the soil fertility, but these effects need to be validated in experimental studies. In particular, few studies have investigated the effects of the combined application of compound fertilizer and biochar on wheat. Hence, in the present study, wheat cultivar Zhongmai 886 was used as the test crop to study the effects of combined application on the wheat yield, soil P content in different growth stages, and P accumulation and utilization in plants. The findings obtained in this study may provide a theoretical and practical basis for improving the absorption and utilization efficiency of nutrients in wheat to promote growth and reduce the use of fertilizer resources.

Materials and Methods

Test site

This study was carried out at the experimental station of the College of Agronomy and Biotechnology, Hebei Normal University of Science and Technology, China (39.70°N, 119.15°E), from 2021 to 2023. The experimental station is located in the temperate continental monsoon climate zone, and the soil type is loam. The soil properties in the 0–20 cm soil layer before planting at the experimental site were as follows: organic matter content of 17.56 g kg−1, total N of 2.30 g kg−1, alkaline hydrolyzable nitrogen of 75.67 mg kg−1, available P of 19.47 mg kg−1, available K of 129.32 mg kg−1, total P of 2.06 g kg−1, and pH of 7.61.

Test materials

The wheat variety used in this study was Zhongmai 886, which was selected by the Chinese Academy of Agricultural Sciences. The biochar type was wheat straw biochar, and the compound fertilizer was a slow-release fertilizer produced by Sakefu with a total content of 43% (N 25%, P2O5 12%, and K2O 6%).

Experimental design

A split plot design was used in the experiment. Biochar was used as the main plot (B) with three treatment levels: 0 quintal (q) ha−1 (B1, no biochar), 75 q ha−1 (B2, low biochar application), and 150 q ha−1 (B3, high biochar application). Compound fertilizer was used as the sub-plot (F) with four treatment levels: 0 q ha−1 (F1, no compound fertilizer), 2.25 q ha−1 (F2, high fertilizer application), 4.50 q ha−1 (F3, low fertilizer application), and 6.75 q ha−1 (F4, conventional fertilization control). Each treatment was replicated three times. The plot area was 6 m2 (3 m × 2 m), with a row spacing of 20 cm. The wheat was sown in strips. Compound fertilizer and biochar were co-applied to the soil in the form of basal fertilizer on October 17, 2021, and winter wheat was sown at the same time. The wheat was harvested on June 18, 2022. In the second year of the experiment (carried out in the same experimental plots as the first year), compound fertilizer was applied on October 7, 2022, and winter wheat was sown at the same time. The wheat was harvested on June 15, 2023.

Sampling and measurement

Grain yield

Samples were collected in triplicate from an area of 1 m2 in each plot during the crop maturity stage. After harvesting, the grains were weighed, threshed, and dried manually, and the water content was measured. Finally, the wheat yield per unit area was calculated at a moisture content of 13%.

Total P and available P contents of soil

In the wheat re-greening stage, jointing stage, booting stage, flowering stage, and maturity stage, soil samples were collected from the 0–20 cm soil layer and mixed evenly. After drying naturally in the air, the soil samples were ground and passed through a 2-mm mesh sieve to remove any soil impurities, before digestion with H2O2–H2SO4. The total P content was determined by using a visible light spectrophotometer, and the available P content was measured by molybdenum–antimony–scandium colorimetry (Bremner, 1996).

Indices related to P accumulation and utilization in plants

Ten plants each with uniform growth were selected in the jointing stage, booting stage, flowering stage, and maturity stage. The aboveground plant parts were divided into the following five parts: leaves, leaf sheaths, stems, glume shell + rachis, and grains. The plant parts were heated in an oven at 105 °C for 30 min and then dried to constant weight at 75 °C. The dry weight of each part was determined and the total was calculated as the accumulated dry matter per plant. The dried samples were crushed using a pulverizer and then digested with H2O2–H2SO4 (Westerman, 1990). The total P content was determined by molybdenum–antimony–scandium spectrophotometry (Thomas, Sheard & Moyer, 1967). The related indices were calculated according to the following formulae:

Plant P accumulation (q ha−1) = plant dry matter accumulation × plant P content;

Partial factor productivity (kg kg−1) = crop yield/P application rate;

Agronomic use efficiency (kg kg−1) = yield in P application area –yield in non-phosphorus application area/P application rate;

Harvest index = (grain P accumulation/total P accumulation at maturity) ×100%.

Data analysis

Excel was used for data collation. SPSS v26 was used to conduct two-factor analysis of variance and post-event analysis. Excel was employed for tabulation and Origin 2024 to draw figures.

Results

Effects of combined application on wheat yield

Figure 1 shows that applying a low level of compound fertilizer significantly reduced the wheat yield, but a slight reduction in compound fertilizer combined with low biochar (B2F3) obtained the highest wheat yields in 2021–2022 (75.06 q ha−1) and 2022–2023 (76.00 q ha−1). In 2021–2022, the wheat yield was significantly higher under B2F3 compared with all other treatments, except for that under B1F4 (conventional fertilization: control). In 2022–2023, the yield was also significantly higher under B2F3 compared with all other treatments, and 10.6% higher compared with that under B1F4.

Figure 1 Effects of different fertilization treatments on wheat yield.

Effects of combined application on total P contents of soil

The compound fertilizer level only had a significant effect on the total soil P content in the re-greening stage. However, the biochar level significantly affected the total soil P contents in the re-greening stage, jointing stage, and maturity stage during 2022–2023. Their interaction had significant effects on the total soil P contents in all stages during 2022–2023, except in the jointing stage and maturity stage (Table 1).

Table 1 Analysis of variance results based on the effects of different fertilization treatments on total phosphorus contents of soil in different wheat growth stages.

Note: * and ** indicate significant effects of variables at P < 0.05 and P < 0.01, respectively, and ns indicates no significant effect (the same applies in the other tables).

	2022–2023 Re-greening stage	2022–2023 Jointing stage	2022–2023 Booting stage	2022–2023 Flowering stage	Maturity stage	
					2022–2023	2021–2022	
Compound fertilizer	*	ns	ns	ns	ns	ns	
Biochar	*	*	ns	ns	ns	*	
C*B	*	ns	*	*	*	ns	

Under B1F1–B1F4 (single application of compound fertilizer), in the jointing stage, the total soil P content increased initially and then declined as the compound fertilizer application rate decreased, and the total soil P content was highest under B1F2, which was 17.4% higher than that under B1F4. During 2021–2022, in the booting stage, flowering stage, and maturity stage, the total soil P content declined as the compound fertilizer application rate decreased, where the contents were highest in the booting stage and flowering stage under B1F3, and in the maturity stage under B1F4 (Fig. 2).

Figure 2 Effects of different fertilization treatments on total phosphorus contents of soil in different wheat growth stages.

Under B1F1–B3F1 (single application of biochar), the total soil P content increased as the biochar application rate increased across all growth stages. The total soil P contents in the jointing stage and two–year maturity stage were highest under B3F1, and 70.6%, 61.9%, and 28.3% higher, respectively, compared with those under B1F1 (P < 0.05). The total soil P contents in the booting stage and flowering stage were highest under B2F1, and 21.2% and 27.4% higher, respectively, compared with those under B1F1 (P < 0.05) (Fig. 2).

Under B2F2–B2F4 and B3F2–B3F4 (compound fertilizer combined with biochar), the total soil P contents in the re-greening stage and jointing stage were highest under B2F2, and 30.9% and 28.6% higher, respectively, compared with those under B1F4 (P < 0.05). The total soil P contents in the booting stage and flowering stage were highest under B2F3, and 28.3% and 28.4% higher, respectively, compared with those under B1F4 (P < 0.05). Compared with B1F4, the lower compound fertilizer application rate combined with low amounts of biochar increased the total soil P contents in the maturity stage during both years. In particular, B3F4 and B2F3 obtained the highest total soil P contents in both years, and they were the most effective treatments (Fig. 2).

Effects of combined application on available P content of soil

The compound fertilizer application rate, biochar amount, and their interaction significantly affected the soil available P contents across all growth periods (Table 2).

Table 2 Analysis of variance results based on the effects of different fertilization treatments on available phosphorus contents of soil in different wheat growth stages.

Note: Asterisks (**) indicate significant effects of variables at P < 0.01.

	2022–2023 Re-greening stage	2022–2023 Jointing stage	2022–2023 Booting stage	2022–2023 Flowering stage	Maturity stage	
					2022–2023	2021–2022	
compound fertilizer	**	**	**	**	**	**	
biochar	**	**	**	**	**	**	
C*B	**	**	**	**	**	**	

Under B1F1–B1F4, in the re-greening stage, jointing stage, and flowering stage, the soil available P contents increased initially and then declined as the compound fertilizer application rate decreased. The soil available P contents in the re-greening stage and jointing stage were highest under B1F2, and 59.0% and 11.4% higher, respectively, compared with those under B1F4 (P < 0.05). In the flowering stage, the soil available P content was highest under B1F3, and 66.9% higher compared with that under B1F4 (P < 0.05). In the booting stage, the soil available P content tended to increase as the compound fertilizer application rate decreased, where the soil available P content was highest under B1F2, and 9.9% higher compared with that under B1F4 (P < 0.05). In 2021–2022, during the maturity stage, the soil available P content decreased initially and then increased as the compound fertilizer application amount decreased. In 2022–2023, during the maturity stage, the soil available P content decreased as the compound fertilizer application amount decreased. Moreover, the soil available P contents were highest under B1F4 in both years (Fig. 3).

Figure 3 Effects of different fertilization treatments on available phosphorus contents of soil in different wheat growth stages.

Under B1F1–B3F1, in the re-greening stage, jointing stage, and flowering stage, the soil available P contents increased initially and then decreased as the biochar application rate increased, where the contents were highest under B2F1, and 23.9%, 53.4%, and 17.5% higher, respectively, compared with those under B1F1 (P < 0.05). In the booting stage, the soil available P content decreased initially and then increased as the biochar application rate increased, where the content was highest under B3F1. During the maturity stage, the soil available P content tended to increase as the biochar application rate increased in both years, where the contents were highest under B3F1, and 34.8% and 47.2% higher, respectively, compared with those under B1F1 (Fig. 3) (P < 0.05).

Under B2F2–B2F4 and B3F2–B3F4, in the re-greening stage, the soil available P contents were highest under B3F4, and 158.3% higher compared with that under B1F4 (P < 0.05). In the jointing stage, booting stage, and flowering stage, the soil available P contents were highest under B3F3, and 61.5%, 28.4%, and 174.7% higher, respectively, compared with those under B1F4 (P < 0.05). Compared with B1F4, reduced compound fertilizer application combined with biochar increased the soil available P contents during the maturity stage in both years, particularly under B3F4. During the maturity stage in 2022–2023, the soil available P content was 74.5% higher under B3F4 compared with that under B1F4 (Fig. 3) (P < 0.05).

Effects of combined application on the accumulation of plant P in different stages

The compound fertilizer application rate, biochar amount, and their interaction significantly affected plant P accumulation in each stage (Table 3).

Table 3 Analysis of variance results based on the effects of different fertilization treatments on accumulation of phosphorus in plants during different wheat growth stages.

Note: Asterisks (**) indicate significant effects of variables at P < 0.01.

	2022–2023 Re-greening stage	2022–2023 Jointing stage	2022–2023 Booting stage	Maturity stage	
				2022–2023	2021–2022	
compound fertilizer	**	**	**	**	**	
biochar	**	**	**	**	**	
C*B	**	**	**	**	**	

Under B1F1–B1F4, in the jointing stage, booting stage, flowering stage, and maturity stage, plant P accumulation increased initially and then decreased as the compound fertilizer application rate decreased. In the jointing stage, booting stage, and flowering stage, plant P accumulation was highest under B1F2, and the values were 20%, 28.6%, and 29.0% higher, respectively, compared with those under B1F4 (P < 0.05). In both years, plant P accumulation was highest under B1F3, and the values were 5.1% and 63.3% higher, respectively, compared with those under B1F4 (Fig. 4) (P < 0.05).

Figure 4 (A–B) Effects of different fertilization treatments on accumulation of phosphorus in plants during different wheat growth stages.

Under B1F1–B3F1, during the jointing stage and maturity stage in 2022–2023, plant P accumulation tended to increase initially as the biochar application rate increased, with the highest values under B3F1 and B2F1, respectively, which were 28.6% and 25.1% higher compared with those under B1F1 (P < 0.05). During the booting stage and maturity stage in 2021–2022, plant P accumulation increased initially and then decreased as the biochar application rate increased, with the highest values under B2F1, which were 76.3% and 5.5% higher, respectively, compared with those under B1F1 (P < 0.05). During the flowering stage, plant P accumulation tended to decrease as the biochar application rate increased, with the highest value under B1F1 (Fig. 4).

Under B2F2–B2F4 and B3F2–B3F4, during the jointing stage, booting stage, and flowering stage, the plant P accumulation amounts were highest under B3F4, B2F3, and B2F2, respectively, and 10%, 88.9%, and 57.6% higher compared with those under B1F4 (P < 0.05). During the maturity stage, B2F3 was one of the optimal treatments over the two years of the experiment, where it significantly increased the plant P accumulation amounts by 93.8% and 13.7%, respectively, compared with B1F4 (Fig. 4).

Effects of combined application on the distribution of plant P in different stages

In the jointing stage (Fig. 5A), the proportion of P was relatively higher in the stems (34.8–57.5%), followed by the leaves (22.9–42.0%), and lowest in the leaf sheaths (19.5–27.8%). The proportions of P in the leaves and sheaths were higher under B2F3, and 5.6% and 21.4% higher, respectively, compared with those under B1F4 (P < 0.05). Moreover, the proportion of P in the stems was highest under B2F2, and 37.6% higher compared with that under B1F4 (P < 0.05).

Figure 5 (A–E) Effects of different fertilization treatments on distribution of phosphorus in plants during different wheat growth stages.

In the booting stage (Fig. 5B), the distribution of P tended to be partially transferred to the glume + rachis. However, the overall proportions of P still followed the order of: leaf (23.2–54.9%) >leaf sheath (16.5–41.8%) >stem (13.7–29.7%) >glume + rachis (5.4–20.7%). The proportion of P in the leaves was highest under B3F3, and 41.1% higher compared with that under B1F4 (P < 0.05). The proportion of P in the leaf sheaths followed the order of: B2F2 > B2F2 > B2F3, and the proportions were 21.5%, 13.1%, and 5.2% higher, respectively, compared with that under B1F4 (P < 0.05). The proportion of P in the stems was highest under B3F4, and 55.5% higher compared with that under B1F4 (P < 0.05). The proportion of P in the glume + rachis was highest under B1F2, and 196.1% higher compared with that under B1F4 (P < 0.05).

The proportion of P in the glume + rachis was higher in the flowering stage than the booting stage (Fig. 5C). The proportions of P in the leaves ranged from 18.7% to 35.6%, and the proportion was highest under B1F4, although not significantly different from that under B2F4. The proportions of P in the leaf sheaths ranged from 15.5–35.8%, where the proportion was highest under B1F2, and 75.5% higher compared with that under B1F4 (P < 0.05). The proportion of P in the stems ranged from 15.8% to 39.1%, where the proportions were highest under: B3F4 >  B2F2 > B2F3, i.e., 134.1%, 114.4%, and 90.4% higher, respectively, compared with that under B1F4 (P < 0.05). The proportion of P in the glume + rachis was highest under B3F1, and 37.9% higher compared with that under B1F4 (P < 0.05).

In the maturity stage (Figs. 5D– 5E), the proportion of P was highest in the grains over the two years of the experiment. Under B3F3, the proportion of P was highest in the leaves during 2021–2022 and in the glume + rachis during 2022–2023, and 9.7% higher in the leaves compared with that under B1F4 during 2021–2022 (P < 0.05). Under B2F4, the proportions of P in the stems and leaves were highest in both years, and significantly higher than those under B1F4 (P < 0.05). During 2021–2022 and 2022–2023, the proportions of P in the leaf sheaths were highest under B1F2 and B3F2, respectively, and significantly higher compared with that under B1F4 (P < 0.05). Under B2F3, the proportion of P was highest in the stems during 2022–2023 and in the glume + rachis during 2021–2022, and they were significantly higher than those under B1F4 (P < 0.05). In addition, the proportions of P in the grains were highest under B2F3 in both years.

Effects of combined application on partial factor productivity, agronomic use efficiency, and harvest index

The compound fertilizer application rate, biochar amount, and their interaction significantly affected the partial factor productivity, agronomic use efficiency, and harvest index (Table 4). The partial factor productivity was significantly higher under B2F2 than all other treatments, and 198.3% higher compared with that under B1F4 (P < 0.05). The agronomic use efficiencies were 139.1% and 121.3% higher under B2F2 and B2F3, respectively, compared with B1F4 (P < 0.05). The harvest index was highest under B1F4, but there were no significant differences between those under B1F3, B3F3, and B3F4.

Table 4 Effects of different fertilization treatments on partial factor productivity, agronomic use efficiency, and harvest index.

Different lowercase letters after the same column indicate significant differences among fertilization treatments (P < 0.05). Note: * and ** indicate significant effects of variables at P < 0.05 and P < 0.01, respectively.

Treatment	2022–2023 Partial productivity (kg kg−1)	2022–2023 Agronomic utilization rate (kg kg−1)	2022–2023 Harvest index (%)	
B1F1	–	–	56.9 ± 0.02de	
B1F2	238.95 ± 0.47c	25.34 ± 0.92c	47.3 ± 0.03f	
B1F3	125.53 ± 0.45f	18.73 ± 1.16d	64.9 ± 0.02ab	
B1F4	84.83 ± 0.95h	13.96 ± 0.89e	65.7 ± 0.01a	
B2F1	–	–	63.8 ± 0.05ab	
B2F2	253.07 ± 0.78a	33.38 ± 0.58a	53.4 ± 0.03e	
B2F3	140.75 ± 0.23d	30.90 ± 0.16b	59.1 ± 0.03cd	
B2F4	87.31 ± 0.54g	14.08 ± 1.17e	59.5 ± 0.05cd	
B3F1	–	–	54.9 ± 0.03e	
B3F2	250.45 ± 0.63b	24.01 ± 0.41c	44.3 ± 0.05f	
B3F3	131.10 ± 0.47e	17.88 ± 0.30d	62.1 ± 0.01abc	
B3F4	83.49 ± 0.35h	8.01 ± 0.91f	61.0 ± 0.02bc	
	Significance test	
Compound fertilizer	**	**	**	
Biochar	**	**	**	
C*B	**	**	**	

Discussion

Effects of combined application on total P and available P contents of soil

Biochar contains the nutrients required for plant growth and its application can help improve the fertility of soil (Hossain et al., 2020). Rangaswami et al. (2020) found that biochar application could increase the total soil P content in wheat fields in a dose-dependent manner. However, in contrast to their findings, the total soil P content did not change significantly as the amount of biochar applied increased in the present study, possibly due to different biochar application rates in the experimental designs, and the contribution of biochar to soil P is also limited by a threshold value and any effect on increases in the soil P content may be lower above this threshold (Zhao et al., 2024). In addition, Cao et al. (2020) showed that combined application increased the total soil P content in corn fields compared with the single application of compound fertilizer. Similar to previous studies, the results obtained in the present study showed that the total soil P contents in wheat fields were higher after biochar application compared with no fertilization and treatment with a single compound fertilizer, and the total soil P contents tended to increase as the amount of biochar increased. Biochar has a large specific surface area and good adsorption performance, and thus it can effectively adsorb and retain nutrients such as N, P, and K in soil and fertilizer, thereby reducing nutrient losses and increasing the nutrient content of soil (Liu et al., 2023).

Previous studies demonstrated that biochar increased the available soil P content in farmland to varying degrees compared with compound fertilizer treatment, and the effect was dose dependent (Arif et al., 2017; Dawerasha et al., 2024; Yamato et al., 2006). In the present study, biochar and compound fertilizer had a significant interactive effect on the available soil P content but it was not due to simple dose superposition, and high biochar treatments had some advantages for maintaining the long-term availability of soil P in the maturity stage. It is possible that increasing the amount of biochar applied promoted the activation of soil P, transforming mineral P and organic P into forms that were readily absorbed by crops, increasing the available P content of the soil (Wabela et al., 2024).

Effects of combined application on P absorption by wheat

NPK fertilizers affect their absorption by wheat, and reasonable application of these three types of fertilizers can promote the absorption of nutrients by wheat. The accumulation of N, P, and K in wheat can reflect their absorption by wheat plants. The capacity for absorption by plants is strong when the nutrient supply is sufficient and these nutrient elements are accumulated at high rates in crops (Yang et al., 2025). In the present study, compared with no fertilizer application, the single application of compound fertilizer increased P accumulation in wheat plants by 6.57–61.52% in different wheat developmental stages. Liu et al. (2024a) found that the combined application of N, P, and K increased P accumulation in plants by 136.45% compared with no fertilization, which is higher than the value obtained in the present study, possibly because any residual P in the soil after previous rice planting was absorbed and utilized by the subsequent wheat crop, thereby increasing P accumulation in wheat plants (Sun et al., 2018). In addition to N, P, and K, the application of biochar affects the absorption of nutrients by plants (Shanmugaraj et al., 2024). The effect of biochar application is not fixed and its influence on promoting plant growth is often strongly related to the fertilization mode and amount applied (Wei et al., 2020). Kang et al. (2014) found that applying compound fertilizer combined with biochar could promote the growth of wheat throughout its whole growing period, and the effect increased as the biochar application amount increased. Iqbal et al. (2019) demonstrated that applying biochar coupled with a reduced amount of compound fertilizer could significantly increase P accumulation in wheat grains. Similar to these previous studies, in the present study, as the amount of compound fertilizer applied decreased, the application of low-dose biochar increased P accumulation in the aboveground plant parts during the late growth stage in wheat, possibly because applying appropriate amounts of N, P, K, and biochar can increase the contents and availability of soil nutrients, thereby enhancing the capacity of plants to absorb nutrients and ultimately enhance crop growth and the yield (Liu et al., 2024b; Sadaf et al., 2017).

In the present study, the peak period for P accumulation in wheat occurred in the flowering stage, followed by significant decreases, which are consistent with the results reported by Sunil et al. (2023) and they may be related to the return of P to the soil (Khan et al., 2018). In addition, our results showed that P mainly accumulated in the leaves of wheat plants in the jointing stage and booting stage, while the accumulation of P in non-leaf organs increased gradually after the flowering stage, and the accumulation of P in grains was highest in the maturity stage. The leaf was the main organ for nutrient absorption in the early stage, but the focus shifted from the leaves to grains in the late stage. The nutrients in each vegetative organ provide the material basis for grain filling (Xu, Hassan & Li, 2023). The results showed that the distribution of P in different organs changed in a regular manner during the growth process under the treatments with reduced compound fertilizer and low biochar (B2F2 and B2F3). In the jointing stage, under B2F2 and B2F3, P was mainly concentrated in the leaves and stems, indicating that fertilization may promote the accumulation of P in photosynthetic organs and supporting structures, thereby enhancing photosynthesis and providing the material basis for tillering (Guo et al., 2021). In the booting stage, under B2F2 and B2F3, the distribution of P began to shift to the glume + rachis, indicating that fertilization may promote the early transfer of P to reproductive organs, which is beneficial for the subsequent development of young ears (Dai et al., 2022). In the flowering stage, under B2F2 and B2F3, the accumulation of P increased significantly in the glume + rachis, reflecting the concentrated supply of P in the ears, thereby ensuring reproductive organ development. Moreover, although the proportion of P in the leaves decreased, it was still maintained at a suitable level to support photosynthesis during later plant growth (Ma et al., 2024). In the maturity stage, P was mainly distributed in the grain, indicating that P plays key roles in grain filling and quality formation. In addition, the residual P in the stem and sheath reflects incomplete P transport after its redistribution in wheat (Wang, Chen & Wu, 2020). In the present study, the proportion of P in the grains varied significantly under different treatments. For example, the allocation of P to the grains in both years was highest under B2F3, indicating that the transport efficiency of P from vegetative organs to the grains was optimized to some extent under B2F3.

Effects of combined application on P utilization and wheat yield

Excessive or insufficient fertilization will reduce the efficiency of P utilization to affect the transfer of P in plants and reduce the yield. Biochar can serve as a carrier of fertilizer and delay the release of nutrients in the soil, thereby reducing nutrient losses and improving the P use efficiency (Zhang et al., 2024; Jia et al., 2024b). Melo et al. (2022) found that compared with single compound fertilizer treatment, the agronomic use efficiency increased significantly by 11.5% using compound fertilizer combined with biochar, which ultimately increased the maize yield. In the present study, compared with conventional single compound fertilizer application, an appropriate compound fertilizer rate combined with biochar significantly increased the P partial factor productivity and agronomic use efficiency by 12.1–65.9% and 21.9–21.4%, respectively, and the yield by 10.6–17.8%, which are similar to the results reported in previous studies (Qodarrohman et al., 2024). Naeem et al. (2017) studied the effects of biochar combined with compound fertilizer on maize growth and the soil properties, and found that the maize grain yield (40% higher than control) and P uptake were significantly greater, thereby increasing the harvest index. Similar to previous studies, reducing the amount of compound fertilizer applied combined with low amounts of biochar increased the harvest index in the present study, whereas excessively high amounts of biochar decreased the harvest index, possibly because applying an excessive amount of biochar can reduce soil aeration and inhibit respiration by the roots to decrease the absorption of nutrients, and thus reduce the harvest index (Jia et al., 2024a).

Conclusion

Applying an appropriate amount of compound fertilizer combined with biochar increased the soil P content, as well as improving the soil physical and chemical properties, thereby enhancing the absorption and utilization of soil P by plants. In addition, improvements were found in the accumulation of P by plants, partial factor productivity, agronomic use efficiency, harvest index, and other indicators to ultimately achieve the aim of high and stable yields. Based on the comprehensive analysis in the present study, the recommended fertilization scheme for the study region is 4.50 q ha−1 compound fertilizer combined with 75 q ha−1 biochar.

Supplemental Information

Supplemental Information 1 Raw data

Additional Information and Declarations

Competing Interests

Author Contributions

Data Availability

The authors declare there are no competing interests.

Yun Li conceived and designed the experiments, performed the experiments, analyzed the data, prepared figures and/or tables, and approved the final draft.

Jiatong Wang conceived and designed the experiments, performed the experiments, analyzed the data, prepared figures and/or tables, and approved the final draft.

Jiahui Lu conceived and designed the experiments, performed the experiments, analyzed the data, prepared figures and/or tables, and approved the final draft.

Yanfei Feng analyzed the data, authored or reviewed drafts of the article, and approved the final draft.

Jingjing Li conceived and designed the experiments, authored or reviewed drafts of the article, and approved the final draft.

Zhenqing Guo conceived and designed the experiments, authored or reviewed drafts of the article, and approved the final draft.

Xiaohu Lin analyzed the data, authored or reviewed drafts of the article, supervision, and approved the final draft.

Yucui Han conceived and designed the experiments, authored or reviewed drafts of the article, and approved the final draft.

The following information was supplied regarding data availability:

The raw measurements are available in the Supplementary File.

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
