# Peer review of "Effects of combined application of compound fertilizer and biochar on absorption and utilization of phosphorus by wheat"

_PeerJ, doi:10.7717/peerj.20308_

## Round 0.1 · original submission · Major Revisions

· Academic Editor

Major Revisions

The authors should address comments of all the reviewers and submit the revised version. When writing the percent change, the author should write only one digit after decimal, for example, 22.6%. Also, please see if the increase below 10% is significant or not, if not, then do not mention it.

**Language Note:** The review process has identified that the English language must be improved. PeerJ can provide language editing services - please contact us at [email protected] for pricing (be sure to provide your manuscript number and title). Alternatively, you should make your own arrangements to improve the language quality and provide details in your response letter. – PeerJ Staff

Reviewer 1 ·

Basic reporting

Major revisions need to be carried out, as comments are given in the attached file. There is incomplete sentences in the file; many sentences are not clear. It should be improved. The discussion section needs improvement.

Experimental design

A better description of the statistical methodology is necessary, and it should meet with the journal’s statistical guidelines.

Validity of the findings

Results section was too long; I suggest shortening it. The authors should show the only significant result in this section.

Findings from the figures should cross-verify.

Figures should be labeled properly.

Additional comments

The attached PDF file mentions numerous corrections and modifications.

Annotated reviews are not available for download in order to protect the identity of reviewers who chose to remain anonymous.

Reviewer 2 ·

Basic reporting

No comment

Experimental design

No comment

Validity of the findings

No comment

Additional comments

I have gone through the manuscript entitled "Effects of combined application of compound fertilizer and biochar on phosphorus absorption and utilization of wheat". The results demonstrated that an appropriate amount of compound fertilizer combined with biochar can increase soil P to varying degrees, as well as improve soil physical and chemical properties, thereby improving the absorption and utilization of plants for P in the soil, the P accumulation of plants. The comprehensive analysis showed that the recommended fertilization scheme was 450 kg•hm-2 compound fertilizer combined with 7500 kg•hm-2 biochar in the studied region. As such the manuscript is well written and relevant information has been added. It may be accepted after minor revisions.

Please find more detailed comments below.
1. Results may be slightly shortened and made more focused in terms of language and presentation
2. More focus should be given to discussion part.
3. In all the figures resolution needs to be improved.
4. Only very relevant references need to be presented.

·

Basic reporting

Authors have taken up an important work to optimise the fertilizer requirement of wheat crop, by supplementing chemical fertilizers with organic products such as biochar, to reduce the deleterious effects to the environment. However, there are several points to be revised before the manuscript may be considered for publication in the esteemed journal PeerJ.
* Introduction is written well, literature well referenced and relevant., however could be revised with inclusion of a clear hypothesis based on the available literature.
* Abstract should be modified to include background, methodology, salient findings, conclusion/perspectives.
* As the manuscript title includes "P absorption and utilization", could authors substantiate the traits used for quantifying P absorption and P utilization per se.

Experimental design

Methods may be revised with more detail on the following aspects:
* LN 104-106: the types of statistical analysis done should be mentioned rather than mere mention of software used.
* Measured traits (yield, tissue P concentration and P uptake, utilization, soil total and available P) are minimal. Could authors provide information on other physiological traits (eg: leaf area index, canopy coverage, photosynthetic rate) or growth indices (eg: crop growth rate, net assimilate rate) that are influenced by biochar+compound fertilizer combination?
* For some traits (eg. LN116, LN134, LN200) data is discussed for two years, while for other traits (eg. LN163, LN 172) year is not mentioned. Why this ambiguity?
* With the availability of two years data, the authors could do a pooled analysis to arrive at conclusive results instead of discussing the trend of individual years.

Validity of the findings

Results section may be revised as suggested below:
* Sub-headings must conform to the Journal guidelines, and also be self explanatory (eg: LN79, LN80)
* Need not repeat the numerical values presented in tables/ graphs. Instead it is better to indicate and compare fold change or relative increase or decrease between treatment combinations
* Mentioning numerical values is taking up a huge chunk of the text which may be avoided.
* LN239: check the relevance of sub-heading as the traits discussed in the following paragraph are PFP, AE and HI.
* PPF, AE and HI presented in Table 1 is for one year or two years?
* Data presented in the tables and graphs suggest significant variation with respect to years 2021-22 and 2022-23. Could authors justify this?

Additional comments

* Pooled analysis of two years data might give a completely different inference of the treatment effects, hence may also require revision of the results and discussion.
* Apart from the above major concerns, rephrasing of sentences as indicated may further improve the manuscript both in terms of scientific and grammatical context. Sentences to rephrase: LN26-27, LN36-37, LN40, LN45, LN51-52, LN77-78, LN79, LN87-90, LN105-106, LN110-111, LN256, LN282,

---

## Round 0.2 · Minor Revisions

· Academic Editor

Minor Revisions

The MS has been carefully revised by the authors however, Reviewer #2 has raised a few pertinent points, which the authors are suggested to address.

Reviewer 1 ·

Basic reporting

no comment

Experimental design

no comment

Validity of the findings

no comment

Additional comments

Authors have included the suggestions and made the corrections.

·

Basic reporting

Authors have carried out the revisions as suggested, but some questions still remain.

Experimental design

Q1: In response to Q13: Data presented in the tables and graphs suggest significant variation with respect to years 2021-22 and 2022-23. Could authors justify this?,
authors have mentioned that biochar application was not repeated in the second year, whereas in Materials and Methods of the revised manuscript, at LN 99, the date of application of biochar has been mentioned for the second year.
Also, if the treatment (combined application of biochar and compound fertiliser) was not repeated for two years, how were the results validated then?

Validity of the findings

Q2: Yield and nutrient status are the only observations recorded in this experiment, others being derived paraments. Hence it is necessary that the same set of treatments are validated atleast for two crop seasons, in order to give fertilizer recommendations.
Authors have mentioned the year of data availability in the tables and figures of the revised MS, but I fail to understand why the data is not available for both the years, and why a pooled analysis is not included to validate the results/ inferences? For eg. Table 4, indices only for one year.
Q3: LN 275-278: "Moreover, this study found that different treatments resulted in the highest content of available P at different growth stages, which may be related to the differences in P absorption capacity of crops at different stages that leads to different fertilizer ratios at different growth stages": There couldn't be more generalised statement than this, which does not require an experimental trial in the first place!
Q5: Existing farmer's practice about fertilizer scheduling for wheat in China may be mentioned to substantiate the need for reduction/ optimisation.
Q6: "In summary, under the experimental conditions of this study, 4.50 q·ha -1 compound fertilizer combined with 75 q·ha -1 biochar is recommended as the optimal fertilization scheme.": whether determination of P absorption and utilization alone sufficient to recommend fertilizer schedule? What about its effect on other essential nutrients?
Q7: Combined application of biochar and mineral fertilizer is not a new concept, there are reports dating back to 2000 or earlier. So, authors may justify the novelty in their study.
https://doi.org/10.1071/CP21095
https://doi.org/10.1007/s00374-013-0845-z
https://doi.org/10.1007/s00374-016-1099-3
https://doi.org/10.3390/agronomy9100623
https://doi.org/10.1007/s00374-012-0746-6
Authors may include observation on traits to validate soil health, to supplement their results.
https://doi.org/10.1016/j.copbio.2024.103198
10.3389/fsufs.2024.1324798

Additional comments

Q8: English writing may be revised carefully to avoid grammatical and typographical errors.

---

## Round 0.3 · accepted · Accept

· Academic Editor

Accept

The authors have addressed all the comments of the reviewers.

·

Basic reporting

English is sufficiently improved, and clear to a wide range of audience.
Literature references are sufficient.

Q1: Lines 43-47 in the revised manuscript is a generalised statement about excess fertiliser application. Please mention in quantitative terms, so that at the end of the study, authors may be able to justify the reduction in chemical fertiliser due to combined application with biochar.

Experimental design

Methods are described with sufficient detail and information to replicate.

Validity of the findings

Q2: It is unclear why authors are not taking up the pooled analysis of traits with what ever data/ variables is available for two years? Pooled analysis may be carried out to get the overall significance of the treatments.

Q3: Table 4: why are these presented only for the second year? PFP, AUR and HI? what about the indices for the year 2021-22?

Q4: Authors mention in their response to Q1 in response file that "there are certain fluctuations in rainfall and temperature between different years. Therefore, the effects of combined application of compound fertiliser and biochar may lead to differences between two years" ; this being the case, can authors justify the statement at lines 341-342 about the fertiliser recommendation based on this "comprehensive analysis in the present study"?

Additional comments

Authors have made considerable revision to the manuscript text, although I do not see any changes with regard to data analysis, or inference of the results.